# Crystal structure of PfRh5, an essential *P. falciparum* ligand for invasion of human erythrocytes

Lin Chen[1,2], Yibin Xu[2,3], Julie Healer[1,2], Jenny K Thompson[1], Brian J Smith[4], Michael C Lawrence[2,3], Alan F Cowman[1,2]*

[1]Division of Infection and Immunity, The Walter and Eliza Hall Institute of Medical Research, Melbourne, Australia; [2]Department of Medical Biology, University of Melbourne, Melbourne, Australia; [3]Division of Structural Biology, The Walter and Eliza Hall Institute of Medical Research, Melbourne, Australia; [4]Department of Chemistry, La Trobe Institute for Molecular Science, La Trobe University, Melbourne, Australia

**Abstract** *Plasmodium falciparum* causes the most severe form of malaria in humans and is responsible for over 700,000 deaths annually. It is an obligate intracellular parasite and invades erythrocytes where it grows in a relatively protected niche. Invasion of erythrocytes is essential for parasite survival and this involves interplay of multiple protein–protein interactions. One of the most important interactions is binding of parasite invasion ligand families EBLs and PfRhs to host receptors on the surface of erythrocytes. PfRh5 is the only essential invasion ligand within the PfRh family and is an important vaccine candidate. PfRh5 binds the host receptor basigin. In this study, we have determined the crystal structure of PfRh5 using diffraction data to 2.18 Å resolution. PfRh5 exhibits a novel fold, comprising nine mostly anti-parallel α-helices encasing an N-terminal β-hairpin, with the overall shape being an elliptical disk. This is the first three-dimensional structure determined for the PfRh family of proteins.

*For correspondence: cowman@wehi.edu.au

**Competing interests:** The authors declare that no competing interests exist.

**Reviewing editor**: Axel Brakhage, Leibniz Institute for Natural Product Research and Infection Biology - Hans Knöll Institute, Germany

## Introduction

*Plasmodium falciparum* is the causative agent of the most severe form of malaria with over 700,000 deaths each year, mostly in sub-Saharan Africa. The asexual blood cycle of this parasite begins with the invasion of human erythrocytes by the merozoite form of *P. falciparum* in a complex multistep process involving a cascade of protein–protein interactions between the parasite and host cell (reviewed in *Cowman and Crabb, 2006*). This process requires members of the reticulocyte binding-like homologues (PfRh or PfRBP) and erythrocyte binding-like (EBL) ligand families.

PfRh5 is a member of the PfRh family and binds specifically to the receptor basigin on the human erythrocyte surface (*Crosnier et al., 2011*). This protein plays an essential role in merozoite invasion (*Baum et al., 2009*) and host tropism of *P. falciparum* (*Wanaguru et al., 2013*). Polymorphisms in PfRh5 can convert a non-virulent *Plasmodium falciparum* parasite into a virulent form upon infection of *Aotus* monkeys, supporting the view that this ligand is a determinant of virulence and host specificity (*Hayton et al., 2008*). PfRh5 has distinct characteristics suggesting that it plays a different role to other members of the family. In particular, PfRh5 is a much smaller protein (~60 kDa compared to the average of ~300 kDa for the family) and lacks a transmembrane region. It forms a complex with the cysteine-rich protein PfRipr during merozoite invasion; the complex is peripherally associated with parasite membranes and is released at the apical end of the merozoite during invasion of the human erythrocyte (*Chen et al., 2011*).

**eLife digest** Malaria is a disease caused by a single-celled parasite called *Plasmodium*, which is transmitted between humans by mosquitoes. It is estimated that 3.4 billion people worldwide live in regions where they are at risk of malaria, and malaria infections cause hundreds of thousands of deaths each year.

When a mosquito carrying *Plasmodium* parasites in its salivary glands bites a human, the parasite is injected into the person's bloodstream with the mosquito's saliva. The parasite then travels through the bloodstream to the liver, where it infects liver cells and multiplies without causing any symptoms for up to 4 weeks. After this period, the parasites break out of each infected liver cell, re-enter the bloodstream, and begin infecting red blood cells. When another mosquito bites the infected individual to feed on their blood, the parasite moves into the mosquito with the red blood cells and the cycle of infection continues.

While prevention and control measures have dramatically reduced the incidence of malaria in some countries, many people in African countries—and especially young children—die from malaria each year. Finding ways to reduce the spread of *Plasmodium* parasites, and in particular *Plasmodium falciparum* (which is responsible for the deadliest type of malaria), is critical for the global effort to control and eliminate this disease. As such, many researchers are trying to gain a better understanding of how the parasite both invades host cells and evades the immune system.

In this study, Chen et al. reveal the high-resolution structure of PfRh5, the protein from *Plasmodium falciparum* that forms a complex with other proteins to allow the parasite to bind to, and invade, red blood cells. This is one of the first three-dimensional structures that have been uncovered for this family of proteins—and reveals that the PfRh5 protein is shaped like an elliptical disk. Solving the structure of PfRh5 is the first step in understanding the role of this protein, and the other protein components, involved in invading red blood cells. These proteins are molecules that could potentially be used to vaccinate people against malaria, and understanding these proteins' functions will help efforts to design vaccines to prevent malarial disease.

Antibodies to PfRh5 can block merozoite invasion, suggesting that it is a potential vaccine candidate (*Douglas et al., 2011*; *Williams et al., 2012*; *Patel et al., 2013*; *Reddy et al., 2014*). This is supported by clinical data showing that antibodies to PfRh5 are associated with protection against malaria, indicating that PfRh5 may be a component of acquired protective immunity (*Chiu et al., 2014*; *Tran et al., 2014*).

To provide a molecular basis for understanding the function of PfRh5, we have determined the crystal structure of PfRh5 using diffraction data to 2.18 Å resolution, the first three-dimensional structure in PfRh protein family. We show that it exhibits a novel fold.

## Results

### Recombinant expression of functional PfRh5

Initially, the full-length 60-kDa protein was expressed in insect cells. Although the recombinant protein was capable of binding red blood cells, it was unstable, with the N-terminal region proteolytically degraded to yield a 48-kDa fragment that had a higher erythrocyte binding affinity than that of the full-length protein (*Figure 1—figure supplement 1*). We determined the N-terminal amino acid sequence by mass spectrometry and re-expressed this region in insect cells to produce a highly stable module of PfRh5 that we denote PfRh5-C (*Figure 1A*). PfRh5-C likely reflects the 45-kDa processed form present in *P. falciparum* and released into culture supernatant during merozoite invasion (*Baum et al., 2009*).

To show that PfRh5-C is functional, we demonstrated that it could bind red blood cells and the receptor human basigin, which is also produced in insect cells (*Figure 1A*). PfRh5-C formed a stable complex with basigin as evidenced by size-exclusion chromatographic analysis. In these experiments, PfRh5-C was incubated with excess basigin and the stable PfRh5-basign complex eluted ahead of free PfRh5 and basigin (*Figure 1B*). The stoichiometry of the complex was shown to be 1:1 by chemical cross-linking (*Figure 1—figure supplement 2*). The binding affinity of the PfRh5-C–basigin interaction

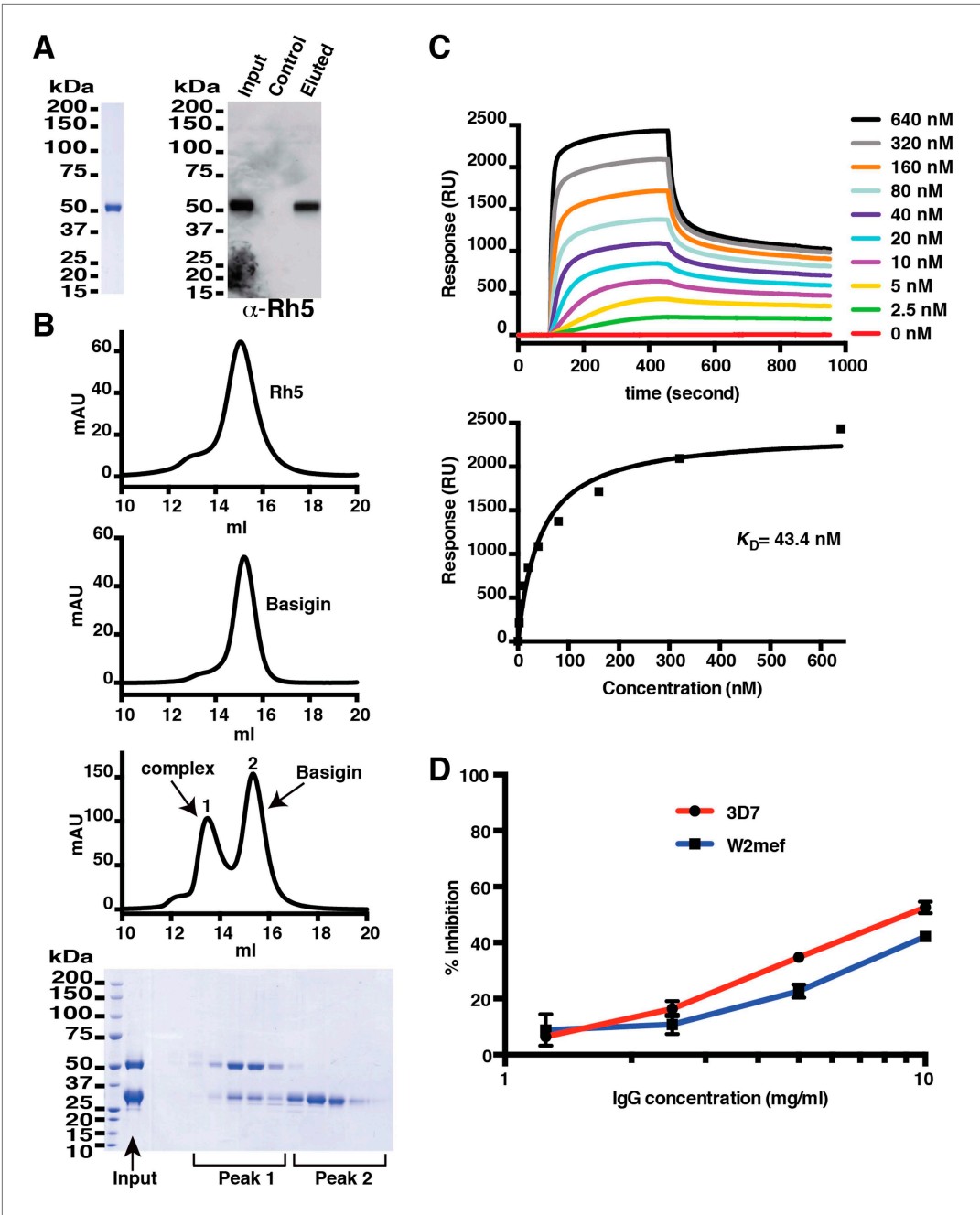

**Figure 1**. Production of functional recombinant PfRh5. (**A**) Purified recombinant PfRh5 was analysed by SDS-PAGE analyses and by erythrocyte binding assays. (**B**) Formation of the PfRh5–basigin complex was monitored by size-exclusion chromatography. The chromatographic profiles are shown for PfRh5 (panel 1), basigin (panel 2), and the PfRh5-basigin complex (panel 3). The fractions eluted from the column in panel 3 were analysed by SDS-PAGE. (**C**) The binding affinity of the recombinant PfRh5 to human basigin was measured by SPR on Biacore 3000 with the basigin coupled to a sensor chip. (**D**) In vitro growth inhibition assays were performed to assess the abilities of the polyclonal antibodies to the recombinant PfRh5 in blocking *P. falciparum* parasite invasion into erythrocytes.

The following figure supplements are available for figure 1:

**Figure supplement 1**. Production of full-length PfRh5.

**Figure supplement 2**. PfRh5 and human basigin form a 1:1 complex.

was determined by surface plasmon resonance (SPR) to be $K_D$ = 43.4 nM (*Figure 1C*). This $K_D$ value is higher than that previously reported (*Crosnier et al., 2011*). We note that while the PfRh5 sample used for the SPR measurement was prepared in monomeric form by gel-filtration chromatography, it is possible that a dynamic equilibrium with oligomeric forms within the sample has contributed to the higher affinity measurement. Antibodies to the recombinant PfRh5-C block growth of 3D7 and W2mef strains of *P. falciparum* at levels comparable to previous studies (*Douglas et al., 2011*; *Bustamante et al., 2013*; *Douglas et al., 2014*; *Reddy et al., 2014*) (*Figure 1D*). Taken together, these data imply that PfRh5-C is functionally competent.

## The crystal structure of PfRh5

PfRh5-C was crystallized and its structure determined through single-wavelength anomalous diffraction (SAD) phasing using iodine-derivatized crystals with subsequent refinement against native diffraction data to a resolution of 2.18 Å (*Figure 2—source data 1*). The shape of PfRh5-C approximates an elliptical disk (*Figure 2*), the core consisting of nine mostly anti-parallel α-helices that encase a small β-hairpin located near the N-terminus (*Figure 2* and *Figure 2—figure supplement 1*). We denote these nine helices α1, α2a, α2b, α3a, α3b, α4, α5, α6, and α7, where the numeric order indicates progression from N- to C-terminus along the polypeptide and the *a* and *b* suffixes indicate that α2a and α2b as well as helices α3a and α3b arise from breaks in the canonical (*i*, *i*+4) hydrogen bonding pattern of longer 'parent helices' α2 and α3, respectively.

Helices α4, α5, α6, and α7 assemble as a triplet-helical coiled-coil domain running the length of the long axis of the molecule. On the opposite side to the α4/α5/α6/α7 coiled-coil domain, helices α1, α2a, and α3b assemble to form a short three-helix bundle and helices α2b and α3a assemble to form a short two-helix coiled-coil domain, these domains being approximately half the length of the α4/α5/α6/α7 coiled-coil domain. The 'absence' of a third helix to the α2b/α3a coiled-coil is necessary to accommodate the small β-hairpin formed by residues 161–175 in the overall tertiary structure. Within each of the three-helical domains, the central pairwise interactions between the constituent helices are overwhelmingly hydrophobic in nature. In contrast, the interactions between the helical domains and the β-hairpin and between the α4/α5/α6/α7 coiled-coil domain and the α1/α2a/α3b bundle domain are of mixed hydrophilicity. Inspection shows these interfaces to be relatively loosely packed.

There are two disulfide bonds within the structure. The first disulphide bond is Cys345–Cys351, located at one apex of the helical bundle. Cys345 lies at the C-terminal end of helix α3b and Cys351 at the N-terminal end of helix α4. The second disulfide bond is Cys224–Cys317; Cys224 lies close to the N-terminus of helix α2b and Cys317 close to the C-terminus of helix α3a, that is, in proximity to 'kink points' of the parent helices α2 and α3. A large loop (residues 240-297), located at the opposite apex of molecule to the Cys345–Cys351 disulphide bond, interconnects helices α2b and α3a (*Figure 2A*). Electron density for residues 258–293 is absent (*Figure 2—figure supplement 1*), indicating partial disorder of this loop within the crystal structure. Two free cysteine residues (Cys203 in α3 and Cys329 in α4) occur within the structure; the side chain of the Cys329 residue is completely buried, whereas that of the Cys203 is only partly exposed (*Figure 2—figure supplement 2*). Burial of these sulfhydryls is consistent with the monomeric nature of the native protein isolated from parasites and of the recombinant protein described here.

A search using DALI (*Holm and Rosenstrom, 2010*) indicates that the PfRh5 fold is novel. The only element of PfRh5-C found to have a structural homologue within the Protein Data Bank (PDB) is the coiled-coil domain formed by helices α5, α6, and α7 (*Figure 2*): this element can be superimposed with a root mean square deviation of all backbone atoms of 3.4 Å on the N-terminal coiled-coil domain (residues 82–226) of the SipB protein of the bacterial type III secretion system (TTSS) of *Salmonella enterica* (PDB entry 3TUL, chain A) (*Barta et al., 2012*) (*Figure 2—figure supplement 3*).

Analyses of molecular surface using DoGSiteScorer (*Volkamer et al., 2012*) detected a number of pockets of dimensions suitable for targeting with small molecules on the surface of the PfRh5 structure. These pockets arise from the relatively loose packing of the four constituent domains of the PfRh5. One of these (*Figure 2—figure supplement 4A*) is lined by residues of mixed hydrophilicity and has a surface area of ~430 Å$^2$, of which ~60% is lipophilic (*Figure 2—figure supplement 4B*). If this pocket is in proximity to the basigin binding site, small molecules targeting it may have the potential to interfere the interaction between the two molecules, either through disrupting the relatively loose packing of the constituent domains of PfRh5 or through steric interference. Alternatively, as PfRh5 functions in complex with PfRipr and at least one further parasite protein (*Chen et al., 2011*), if this pocket is

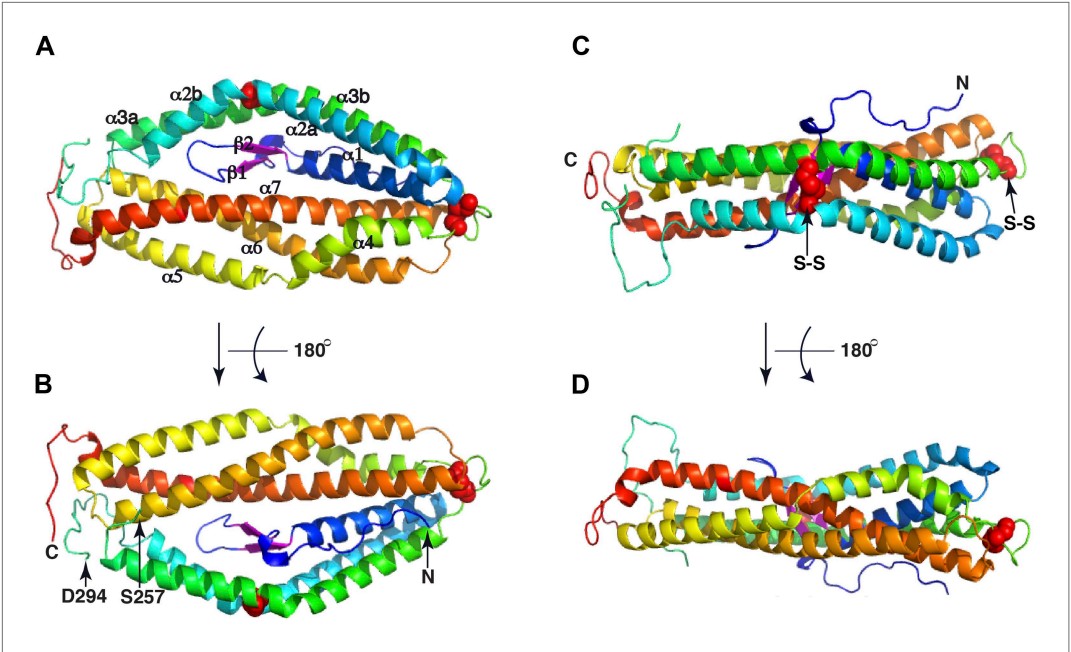

**Figure 2**. The crystal structure of PfRh5. (**A**) Ribbon representation of the PfRh5 structure. Color scheme is rainbow (N-terminus: blue; C-terminus: red) except for the β-hairpin that is colored magenta for clarity. The cysteine residues that form disulfide bridges (Cys345–Cys351 and Cys224–Cys317) are shown as spheres. Helices α4, α5, α6, and α7 assemble as a triplet-helical coiled-coil domain running the length of the long axis of the molecule, helices α1, α2a, and α3b assemble to form a short triplet-helical bundle and helices α2b and α3a assemble to form a short two-helix coiled coil. (**B**) Ribbon representation of the PfRh5 structure viewed after a 180° rotation relative to that in (**A**). The residues between Ser257 and Asp294 are disordered. (**C**) Ribbon representation of the PfRh5 structure viewed from the side with the C-terminus on the left. The disulfide bridges are indicated with arrows. (**D**) Ribbon representation of the PfRh5 structure viewed after a 180°-rotation relative to that in (**C**).

The following source data and figure supplements are available for figure 2:

**Source data 1**. Data collection and refinement statistics of Rh5 and Rh5_KI.

**Figure supplement 1**. The secondary structure of PfRh5.

**Figure supplement 2**. Two free cysteine residues (Cys203 on α2a and Cys329 on α3b) within the crystal structure of PfRh5.

**Figure supplement 3**. Superimposition of the PfRh5 structure with N-terminal coiled-coil domain of SipB.

**Figure supplement 4**. A unique pocket on the surface of the PfRh5 molecule.

involved in binding these partner/s, a small molecule targeting this pocket may likewise interfere with the complex formation and therefore ultimately with its function.

To explore the structure–function relationship of PfRh5, we examined the relevance of the disulphide bonds and cysteine residues for the function of PfRh5 by reducing and alkylating PfRh5-C followed by measuring the binding affinity of the modified protein to basigin using surface plasmon resonance (**Figure 3A**). The reduced and alkylated PfRh5-C has an affinity for basigin with $K_D$ = 127 nM. A threefold reduction in affinity as compared to untreated PfRh5-C ($K_D$ = 43.4 nM) was consistent with the two disulphide-bonds being important for the stability of the overall fold of the protein rather than being directly involved in basigin binding.

We also investigated whether the disordered region of 35 amino acids (Glu258–Asn293) (**Figure 2—figure supplement 1**) is involved in the binding of PfRh5 to human basigin. A mutant form of PfRh5-C lacking amino acids Asp261–Asn289 (**Figure 2—figure supplement 1**) was produced in insect cells; its

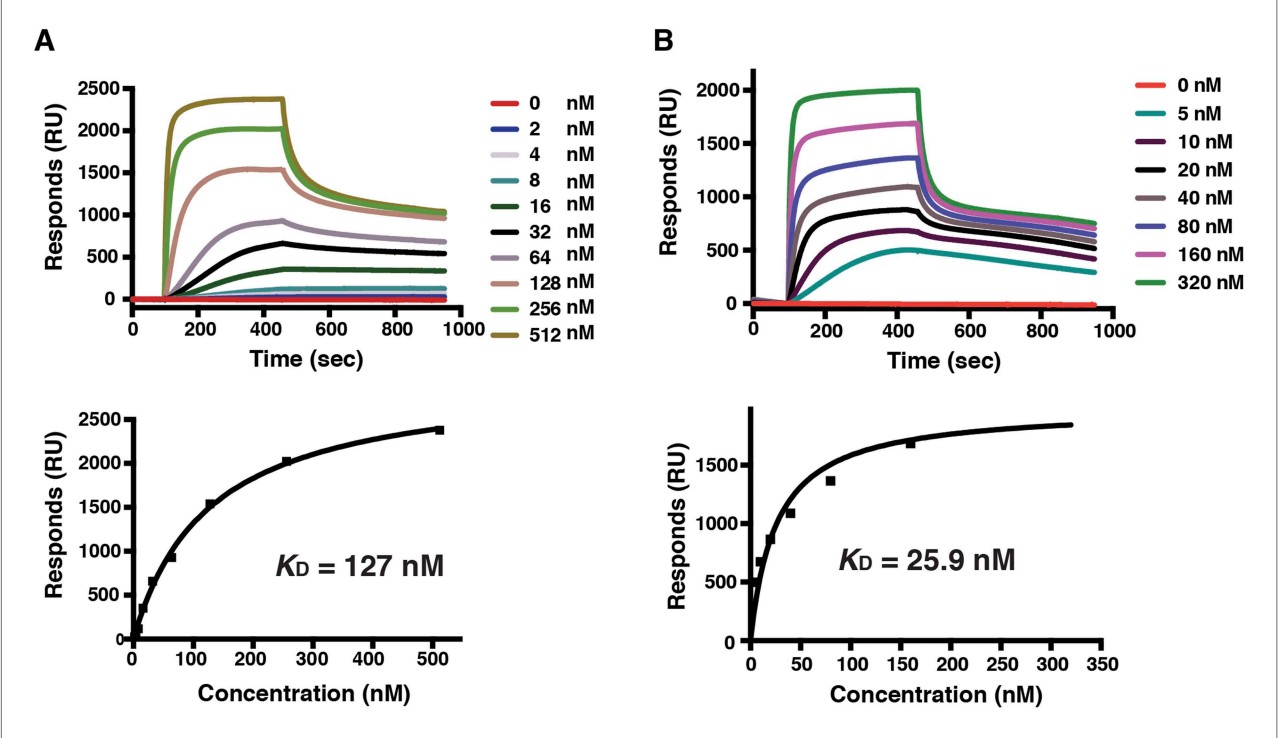

**Figure 3**. Binding of mutant PfRh5 to basigin. (**A**) The binding affinity of the reduced and alkylated PfRh5 to basigin measured by SPR. (**B**) The binding affinity of the PfRh5 mutant, in which the disordered loop has been deleted, to basigin measured by SPR.

binding affinity for basigin was determined by SPR to be 25.9 nM (*Figure 3B*), that is, comparable to that of the non-mutated PfRh5. These data suggest that the disordered region (Glu258–Asn293) is not involved in receptor binding.

## Discussion

Invasion of *P. falciparum* merozoite into human erythrocytes involves several ligand–receptor interactions including the PfRh family of proteins, which are important in binding to and identifying the appropriate host cell for invasion (reviewed in *Cowman and Crabb, 2006*). Whilst PfRh5 is a member of the PfRh family, it appears to have distinct functions to other family members and it plays an essential role for invasion (*Baum et al., 2009*; *Crosnier et al., 2011*). To provide a structural basis for understanding the function of this protein, we determined its three-dimensional structure, the first crystal structure in the PfRh protein family. PfRh5 exhibits a novel fold comprising three-helical domains surrounding a small β-hairpin.

PfRh5 appears to function as a multi-protein complex. We have previously identified one partner, PfRipr, for the complex (*Chen et al., 2011*). The PfRh5–PfRipr complex is essential for merozoite invasion as the genes encoding both proteins cannot be disrupted and antibodies to both inhibit this process (*Baum et al., 2009*; *Chen et al., 2011*). Consequently, both proteins are potential vaccine candidates and worth consideration for a combination vaccine because specific antibodies to both molecules would inhibit the same functional process and likely to be at least additive and potentially synergistic.

The structural similarity of the larger PfRh5 coiled-coil domain with the N-terminal coiled-coil domain of SipB may potentially provide an indication of the function of the complex. SipB forms part of the Salmonella type III secretion system (TTSS) that is responsible for transport of bacterial effector proteins across the host cell membrane. Currently, there is no direct evidence that the PfRh5–PfRipr complex is involved in transport of proteins or molecules across the erythrocyte membrane during invasion. Nevertheless, it is interesting that *P. falciparum* does inject proteins, including members of the RON complex, into the erythrocyte during merozoite invasion and these proteins are required for

formation of the tight junction that bring the parasite membrane and host membrane together in a tight interaction through binding of RON2 to apical membrane antigen-1 (AMA-1) (*Narum et al., 2008*; *Richard et al., 2010*; *Riglar et al., 2011*). By analogy to the bacterial type III secretion system, the PfRh5–PfRipr complex may play a role in transfer of components such as the RON complex to the host cell during merozoite invasion.

In summary, this work has elucidated the structure of PfRh5 and may provide a model for the remainder of the PfRh family members. During the review process of this manuscript, a study was published reporting the structure of the Rh5–basigin complex (*Wright et al., 2014*). In contrast, our work describes the structure of Rh5 not bound to the receptor basigin. Comparison of the Rh5 structure in the bound and unbound state shows that there are no changes in the core structure on binding of receptor. A pocket identified on the surface of PfRh5 may provide an opportunity for development of a new anti-malaria drug. Structural similarity with SipB has provided the tantalizing possibility that PfRh5–PfRipr complex may play a similar role to the TTSS system of bacteria that secrete effector proteins into the host cell. However, further evidence will be required to support such speculation.

# Materials and methods

## Protein expression and purification

A synthetic gene encoding *Plasmodium falciparum* (3D7) full-length mature PfRh5 (residues 24–526), PfRh5-C (residues 127–526), or its mutant was inserted into insect/mammalian cell expression vector pgpHFT (*Xu et al., 2010*) using Kpn I and Xho I sites to produce pgpHFT-PfRh5. The pgpHFT-PfRh5 was then co-transfected with FlashBAC (Oxford Expression Technologies) into Sf21 insect cells as per supplier's manual. The seed virus was amplified to obtain high-titer viral stocks, which were then used to infect Hi5 cells grown in express Five SFM medium (Life Technologies Pty Ltd, Australia) supplemented with 1 mM glutamine. The supernatant containing the secreted recombinant protein was harvested, centrifuged, and passed over anti-FLAG M2 agarose (Sigma-Aldrich, Australia) column. After extensive washing, bound proteins were eluted from the column with the FLAG peptide at a concentration of 100 μg/ml, concentrated and further purified by size-exclusion chromatography with a Superdex 200 column (GL 10/300, GE Healthcare, Australia) in 50 mM Tris, 100 mM NaCl, pH 8.5. For crystallization of PfRh5, the tandem 6xHis and FLAG tags were removed by digestion with a TEV protease and the pure protein was recovered by Ni-resin and/or size-exclusion chromatography purification. Human basigin isoform 2 (BSG-S) (*Crosnier et al., 2011*) was also expressed in insect cells and purified as described for PfRh5.

## Crystallization and structure determination

Crystallization trials were performed in sitting drop within a 96-well format at 8°C. Crystals were obtained from drops containing 8–12% PEG3350 or PEG4000 and 0.2 M DL-malate-imidazole, pH 6.5-7.5. X-ray diffraction data were collected on beamline MX2 at the Australian Synchrotron. For the phase determination, derivative crystals were prepared by quick-soaking native crystals in potassium iodide (KI) solutions prepared in cryoprotection solution for 1–5 min and data collected at the K-edge of 1.55 Å.

Diffraction data were processed and scaled with XDS (*Kabsch, 2010*). Diffraction data were included to a maximum resolution of 2.18 Å based on significance of the $CC_{1/2}$ criterion at the p = 0.001 level of significance (*Karplus and Diederichs, 2012*). Three iodine binding sites were found using SOLVE (*Terwilliger and Berendzen, 1999*); phases were then improved using RESOLVE (*Terwilliger, 2000*). The initial model generated by the RESOLVE autobuild utility comprised only a partial set of the nine helices. Model building and refinement then continued with PHENIX (*Adams et al., 2010*), with automated building and morphing routines leading to a model comprising approximately 50% of the PfRh5 sequence. Further rounds of refinement and manual rebuilding were undertaken using REFMAC5 (*Vagin et al., 2004*) and COOT (*Emsley et al., 2010*). The final refinement was undertaken with AutoBUSTER (v2.10.0) (*Bricogne et al., 2011*). Data processing and refinement statistics are in *Figure 2—source data 1*.

## Antibody production and growth inhibition assays

Rabbits were immunised three times with 200 μg PfRh5 in Freund's adjuvant. IgG was purified from serum, concentrated and dialysed against RPMI-Hepes for growth inhibition assays. One cycle growth inhibition assay was performed as described (*Healer et al., 2013*). Serial dilutions of IgG in RPMI-HEPES,

starting at 10 mg/ml were added to *P. falciparum*-infected RBC (3D7 and W2mef) at a parasitaemia of 0.5%. Parasitaemia was counted after 48 hr and specific growth inhibition calculated relative to parasites grown in non-immune IgG.

## Erythrocyte binding assay, complex preparation and affinity measurement

Erythrocyte binding assays were performed as described previously (*Triglia et al., 2001*). To prepare and analyse the PfRh5/basigin complex, two anti-flag affinity beads purified proteins were mixed, incubated at 4°C overnight and loaded to a Superdex 200 size-exclusion chromatography column in 50 mM Tris, 150 mM NaCl, pH 8.5 or 25 mM HEPES, 150 mM NaCl, pH 7.2. The complex eluted from the column was collected for analysis.

Affinity ($K_D$) measurements were performed at room temperature on a Biacore 3000 Biosensor with HBS (10 mM HEPES pH 7.2, 150 mM NaCl, 3.4 mM EDTA, 0.005% Tween 20) as the running buffer (*Chen et al., 2011*). The human basigin was immobilized onto a CM5 sensorchip using amine-coupling (EDC/NHS) chemistry. PfRh5 or its mutants (purified by anti-FLAG M2 agarose beads followed by gel-filtration chromatography) were injected at 20 µl/min into the sensorchip containing a channel immobilized with basigin. A blank channel was used as control. After each injection, the chip was regenerated with 2 M NaCl supplemented with 3 mM NaOH in the running buffer, followed by two washes with the running buffer. Affinity ($K_D$) was derived from sensorgrams, following subtraction of baseline responses, using BIA evaluation software (version 4.1: Biacore Life Sciences, GE Healthcare, Australia).

## Reduction and alkylation of PfRh5

To an aliquot of 50 µl at 1 mg/ml PfRh5 was added DTT to a final concentration of 2 mM and incubated at room temperature for 2 hr. Iodoacetamide was then added to a final concentration of 10 mM, incubated at room temperature for 2 hr and then left at 4°C overnight. For SPR experiment, the excess DTT and iodoacetamide were removed by a desalting column equilibrated and eluted with HEPES buffer, pH 7.4 containing 150 mM NaCl.

## Chemical crosslinking

For cross-linking of PfRh5 to Basigin, to approximately 50 µg of purified PfRh5/Basigin complex in 25 mM MES, 150 mM NaCl, pH 7.4 was added 1-ethyl-3-(dimethylaminopropyl) carbodiimide (EDC) and *N*-hydroxysuccinimide (NHS) to a final concentration of 2 mM. The reaction was allowed to occur at room temperature for 30 min before quenching with 100 mM Tris, pH 8.0. The cross-linked sample was then analysed on a SDS-PAGE gel.

## Acknowledgements

We thank the Victorian Red Cross Blood Bank for supply of blood. Docking calculations were supported by computational resources at the VLSCI.

## Additional information

### Funding

| Funder | Grant reference number | Author |
|---|---|---|
| National Health and Medical Research Council | 637406 | Alan F Cowman |
| Howard Hughes Medical Institute | 55007645 | Alan F Cowman |
| Department of Health, Victorian State Government | OIS, NHMRC, IRISS Grants | Michael C Lawrence, Alan F Cowman |
| PATH | Malaria Vaccine Initiative 07608-COL | Julie Healer, Alan F Cowman |
| United States Agency for International Development | 07608-COL | Julie Healer, Alan F Cowman |

The funders had no role in study design, data collection and interpretation, or the decision to submit the work for publication.

## Author contributions

LC, Conception and design, Acquisition of data, Analysis and interpretation of data, Drafting or revising the article, Contributed unpublished essential data or reagents; YX, JKT, BJS, Acquisition of data, Analysis and interpretation of data; JH, Conception and design, Acquisition of data, Analysis and interpretation of data; MCL, Analysis and interpretation of data, Drafting or revising the article, Contributed unpublished essential data or reagents; AFC, Conception and design, Analysis and interpretation of data, Drafting or revising the article

## Additional files

### Major dataset

The following previously published dataset was used:

| Author(s) | Year | Dataset title | Dataset ID and/or URL | Database, license, and accessibility information |
| --- | --- | --- | --- | --- |
| Barta ML, Dickenson NE, Patil M, Keightley A, Wyckoff GJ, Picking WD, Picking WL, Geisbrecht BV | 2012 | Crystal structure of N-terminal region of Type III Secretion Major Translocator SipB (residues 82-226) | http://www.rcsb.org/pdb/explore/explore.do?structureId=3tul | Publicly available at RCSB Protein Data Bank. |

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
