## [Decision Letter]

Thank you for sending your work entitled “Crystal structure of PfRh5, an essential *P. falciparum* ligand for invasion of human erythrocytes” for consideration at *eLife*. Your article has been favorably evaluated by Richard Losick (Senior editor), a Reviewing editor, and 2 reviewers.

The Reviewing editor and the reviewers discussed their comments before we reached this decision, and the Reviewing editor has assembled the following comments to help you prepare a revised submission.

Your manuscript presents the X-ray crystal structure of a *Plasmodium falciparum* reticulocyte binding-like homologue, PfRh5. It reports the purification, molecular manipulation, binding affinities and X-ray crystal structure of PfRh5. It further discusses the implications of the molecular detail of PfRh5 to future efforts in vaccine development, protein-protein interference and speculates on the role of PfRh5 protein complexes.

Although we think the manuscript contains valuable data, a problem is the recently published structure by Wright et al. (Nature 2014, August 17). However, because your manuscript was submitted before the release of the competing paper we offer to publish yours as long as the following criteria are met:

1) The speculative aspects, that is those which are not supported by experiments (or at least only negative data), are removed from the manuscript. This would certainly be the simulated docking (Figure 3) and possibly the surface description (Figure 4).

2) If you wish to publish the BIAcore data (Figure 1) then there needs to be an acknowledgement in the main text that the RH5 preparation used for these studies contains oligomeric forms which leads to an over estimation of the KD. This will stop any hard-to-resolve inconsistencies appearing in the literature since their value of 43nM differs significantly from the original measurement (1.1uM Crosnier et al.) and the recent Nature structure paper (1.3uM Wright et al.).

More specifically, the affinity of the Rh5-basigin interaction reported in your manuscript is significantly stronger to that reported in the original Crosnier et al. paper. The reason for this is apparent from the sensorgrams in Figure 1 which shows the clear presence of a mixture of different Rh5 forms; probably multimeric Rh5 material in the preparation. While this doesn't affect the conclusions of the paper, it does affect the reported affinity measurements.

3) One technical concern you need to comment on is the resolution limit of the structure presented. Your manuscript states that the X-ray crystal structure is 2.18 Angstrom; however, the statistics in Table 1 do not support this. The values of the I/sigI (0.52) and CC(1/2) (0.164) in the outer shell are well below what is generally considered acceptable (or recommended) limits (I/sigI > 1 in conjunction with a CC1/2 of > 0.5). Combined with the high Rmerge in the outer shell (530.8), we would question the selection of 2.2 Angstrom and argue that it is not likely to be that high. Can you justify the choice of resolution limit or do you need to modify the structure accordingly?

We support the stance that the paper should still be considered despite a competing paper being published subsequent to submission, because the RH5 structure still has high value.

---

## [Author Response]

*1) The speculative aspects, that is those which are not supported by experiments (or at least only negative data), are removed from the manuscript. This would certainly be the simulated docking (*Figure 3*) and possibly the surface description (Figure 4)*.

Figure 3 (Simulated docking) and its relevant text have now been removed. Figure 4 has been moved to a figure supplement and its relevant text is now under the Results section: The crystal structure of PfRh5.

*2) If you wish to publish the BIAcore data (*Figure 1*) then there needs to be an acknowledgement in the main text that the RH5 preparation used for these studies contains oligomeric forms which leads to an over estimation of the KD. This will stop any hard-to-resolve inconsistencies appearing in the literature since their value of 43nM differs significantly from the original measurement (1.1uM Crosnier et al.) and the recent Nature structure paper (1.3uM Wright et al.)*.

*More specifically, the affinity of the Rh5-basigin interaction reported in your manuscript is significantly stronger to that reported in the original Crosnier et al. paper. The reason for this is apparent from the sensorgrams in*
Figure 1
*which shows the clear presence of a mixture of different Rh5 forms; probably multimeric Rh5 material in the preparation. While this doesn't affect the conclusions of the paper, it does affect the reported affinity measurements*.

A sentence has now been added into the main text to address this issue as has been requested.

We agree that oligomeric forms may be contributing to the higher affinity that we report compared to other studies such as Crosnier et al., Nature 2011: 480: 534-537. We have added the following sentence to note this:

‘The binding affinity of the PfRh5-C–basigin interaction was determined by surface plasmon resonance (SPR) to be *K*_D_= 43.4 nM (Figure 1). This *K*_D_ value is higher than that previously reported. We note that while the PfRh5 sample used for the SPR measurement was prepared in monomeric form by gel-filtration chromatography, it is possible that a dynamic equilibrium with oligomeric forms within the sample has contributed to the higher affinity measurement.”

*3) One technical concern you need to comment on is the resolution limit of the structure presented. Your manuscript states that the X-ray crystal structure is 2.18 Angstrom; however, the statistics in Table 1 do not support this. The values of the I/sigI (0.52) and CC(1/2) (0.164) in the outer shell are well below what is generally considered acceptable (or recommended) limits (I/sigI > 1 in conjunction with a CC1/2 of > 0.5)*. *Combined with the high Rmerge in the outer shell (530.8), we would question the selection of 2.2 Angstrom and argue that it is not likely to be that high. Can you justify the choice of resolution limit or do you need to modify the structure accordingly?*

Our resolution cutoff has been set based on significance of the CC(1/2) statistic at the p=0.001 level of statistical significance as reported by the XDS package. Resolution cut-offs have been the subject of debate in the X-ray community for decades and we have followed the recently expressed sentiment by experts in the field (PR Evans & GN Murshudov (2013) How good are my data and what is the resolution? Acta Cryst. D69, 1204–1214): “... it seems that changing the resolution cutoff over a considerable range (e.g. from 2.2 to 1.9 A) makes only a small difference, so the exact cutoff point is not a question to agonize over, but it seems sensible to set a generous limit so as not to exclude data containing real (if weak) information.”

We also note that it is now generally accepted that Rmerge is a useless criterion for deciding on resolution, given that it is a function of data multiplicity and tends to infinity as the data become weaker. Nevertheless, to make the cutoff issue clear to the reader, we have inserted both in the Methods section and in the diffraction data table an explicit statement about how the resolution cutoff has been set.

In line with this approach, we have also take care throughout the manuscript to state that the structure was “refined against native data to 2.18 Angstroms resolution”, rather than to make a statement of the form that “the resolution of the structure is 2.18 Angstroms”.